# MCFEND: A Multi-source Benchmark Dataset for Chinese Fake News Detection

## ABSTRACT

The prevalence of fake news across various online sources can have significant influence to the public. Existing Chinese fake news detection datasets are limited to the news sourced from Weibo solely. However, fake news that originates from multiple sources exhibits diversity across various aspects, including its content and social context. Methods trained on data from such a single news source can be hardly applicable to the real-world scenarios. Our pilot experiment demonstrates that the macro F1 score of the state-of-the-art method trained on the largest Chinese fake news detection dataset *Weibo-21* so far, drops from 0.98 to 0.47 when changing the test data from *Weibo-21* to multi-source data, failing to identify 35.34% of the multi-source fake news. To address this limitation, we construct the first multi-source benchmark dataset for Chinese fake news detection, termed MCFEND, which contains news collected from diverse sources, such as social platforms, messaging apps, and traditional online news outlets, and fact-checked through 14 authoritative fact-checking agencies. In addition, various established Chinese fake news detection methods are thoroughly evaluated on our proposed dataset, including the state-of-the-art approaches, in both the *cross-source* and *multi-source* scenarios. MCFEND contributes to the field of fake news detection by aiming at a benchmark to evaluate and advance Chinese fake news detection approaches in real-world scenarios.

## CCS CONCEPTS

• **Information systems** → **Data mining**; • **General and reference** → **Evaluation**; • **Computing methodologies** → **Natural language processing**.

## KEYWORDS

Multi-source Benchmark Dataset, Chinese Fake News Detection, Cross-source Evaluation, Multi-source Evaluation

## 1 INTRODUCTION

It has been prevalent for people to obtain news through various online sources, such as social platforms and news websites. At the same time, such sources are efficient media for spreading fake news. For instance, the latest Weibo's annual report on fake news [32] reported that Weibo's official fact-checking agency identified 82,274 pieces of fake news in the last year. Given the potential devastating consequences of fake news on both individuals and society, fake news detection has become an urgent and essential task that needs to be addressed [1, 8, 15, 30]. Therefore, several Chinese fake news detection dataset have been constructed to promote the development of Chinese fake news detection [8–11, 15, 34, 36].

The existing Chinese fake news detection datasets are limited to Weibo as the only source of both true and fake news. However, in the real world, fake news emerges from multiple sources, such

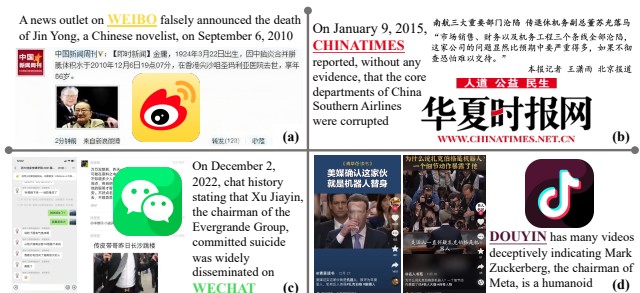

**Figure 1: An example of four pieces of fake news originating from four different Chinese news sources, including Weibo (a popular social platform), China Times (an online news outlet), Wechat (a messaging app), and Douyin (a social platform). Each piece of fake news showcases different characteristics across various aspects, such as content, topics, publishing methods, linguistic styles, etc.**

as social platforms, messaging apps, traditional online news outlets, etc. Fake news from different sources is characterized by its diversity in terms of content, topics, publishing methods, and the utilization of sophisticated linguistic styles intended to mimic real news [1, 16, 18, 25, 31]. For example, Fig. 1 shows four instances of fake news respectively sourced from four distinct news sources, each exemplifying different characteristics. Therefore, we argue that the existing Weibo based Chinese fake news detection datasets fail to capture the above data diversity and can lead to several weaknesses regarding machine learning (ML) based fake news detection, including but not limited to the robustness to intricately crafted fake news and the generalization to fake news from other sources [1, 8, 15, 18, 28].

**Pilot Experiment**. To verify the extent of such limitations, we collected 817 pieces of fake news, verified between Jan. 2015 and Mar. 2023, from the China Internet Joint Rumor Refuting Platform[1], a government-backed fact-checking agency supported by authoritative experts and various government departments. The agency covers fake news originating from a wide variety of sources, including but not limited to Douyin[2], Wechat[3], TouTiao[4], Zhihu[5], Weibo[6], etc. We trained the state-of-the-art model using BERT-EMO [36] on the *Weibo-21* dataset [15]. The model demonstrated strong performance with F1 scores of 98.00 on the *Weibo-21*, 93.20 on the

---

[1]https://www.piyao.org.cn/
[2]https://www.douyin.com/
[3]https://www.wechatapp.com/
[4]https://www.toutiao.com/
[5]https://www.zhihu.com/
[6]https://m.weibo.cn/

*Weibo-20* [36] dataset and 90.80 on the *Weibo-16* dataset [10].[7] Nevertheless, when applied to detect fake news collected from all the diverse sources on the platform, the model failed to identify 35.34% of fake news. Its macro F1 score drops to 47.01 by 52.03%. The results may be attributed to two key factors. First, during the training phase, ML models lack exposure to a diverse range of data from various sources, which leads to overfitting to specific characteristics of Weibo fake news. Consequently, this limits their capability to generalize and effectively identify fake news from different sources. Further, during the testing phase, these models are evaluated using Weibo data exclusively, overlooking a comprehensive assessment of their performance across different news sources. Given that fake news emerges from multiple sources, the applicability of models trained and tested on existing datasets may not extend to real-world scenarios. The results validate the limitations of current Chinese fake news detection datasets. Therefore, it is imperative to construct a comprehensive dataset that includes news from diverse sources.

To bridge this gap, we constructed the first Multi-source benchmark dataset for Chinese FakE News Detection (MCFEND), which contains 23,974 pieces of authoritatively verified Chinese news from 14 fact-checking agencies covering numerous news sources.[8] These fact-checking agencies are divided into three distinct groups. The first group encompasses nine Chinese fact-checking agencies that have been verified by experts as active and authoritative. The second group utilizes three existing annotated English fake news detection datasets. Specifically, for an English news piece paired with its corresponding authenticity label from an existing English fake news dataset, we employ a cross-lingual identical news retrieval method to collect its Chinese equivalent while retaining its original label. The third group consists of the Weibo's official fact-checking agency exclusively, Weibo Community Management Center. For news collected from this group, we directly utilized news data from the *Webo-21* dataset [15]. Furthermore, we conducted comprehensive evaluations on eight established baseline models for Chinese fake news detection, including the state-of-the-art methods, under cross-source and multi-source scenarios using MCFEND dataset. The experimental results characterize the challenge of accurately spotting fake news from different sources that the dataset presents.

Our contributions are summarized as follows. (1) We constructed the initial multi-source Chinese fake news detection (MCFEND) dataset[9], which comprises multi-modal content and social context of 23,974 real-world Chinese news pieces collected from 14 authoritative fact-checking agencies in three distinct groups. Besides, to the best of our knowledge, MCFEND is also the largest open-sourced Chinese fake news detection dataset, being at least 2.63 times larger than existing datasets. The dataset aims to benchmark the evaluations of Chinese fake news detection methods in real-world scenarios, where news originated from diverse sources, and encourage further research in this field. (2) We conducted cross-source and multi-source evaluations comprehensively on eight established baseline models for Chinese fake news detection, including state-of-the-art methods. Our experimental results reveal that the models trained on existing datasets is not applicable in the real-world scenarios. Incorporating multi-source data is necessary which results in substantial improvements in models' robustness.

## 2 RELATED WORK

Fake news detection, also referred to as false news detection, and related to information credibility evaluation, is commonly defined as a binary classification task [3, 8, 17, 18]. In this context, the output space is defined as $\mathcal{Y} = \{0, 1\}$, indicating whether a given piece of news is fake (1) or not (0). The input space, denoted as $\mathcal{X}$, encompasses multi-dimensional information, including news content and social context. Formally, let $\mathcal{D} = \{(x_i, y_i)\}_{i=1}^n$ represent a collection of $n$ news with annotated labels. Given $\mathcal{X}$, the objective of fake news detection models is to learn a mapping function $\phi : \mathcal{X} \to \mathcal{Y}$. Here, $\mathcal{X} = \{x\}_{i=1}^n$ denotes the multi-dimensional information for individual news pieces, and $\mathcal{Y} = \{y_i\}_{i=1}^n \subset \{0, 1\}^n$ represents the corresponding authenticity labels.

Numerous datasets have been constructed to address fake news detection. Representative English fake news detection datasets, such as *BuzzFace* [22], *LIAR* [29], *FakeNewsNet* [24], *PHEME* [37], *KaggleFakeNews* [21], *FakeNewsCorpus* [16], and *FakeHealth* [6], collect English news from social platforms like Twitter and Facebook, as well as fact-checking websites, such as BuzzFeed, PolitiFact, and NewsGuard.[10] A few Chinese fake news detection datasets have been constructed as well. For instance, Ma et al. introduced the *Weibo-16* dataset [10], collected from the Chinese social platform Weibo. This dataset contains verified fake news sourced from the Weibo Community Management Center[11], the official fact-checking agency for posts on Weibo. Real news were collected from regular posts that were not categorized as fake. While *Weibo-16* focuses exclusively on textual data, Jin et al. [11] later introduced *Media-Weibo*, the first multi-modal dataset for detecting Chinese fake news. *Media-Weibo* includes textual content, user profiles, and supplementary images for each post. Zhang et al. [36] then extended the *Media-Weibo* and constructed *Weibo-20*, which enriched the dataset by adding 850 real news which authenticated by NewsVerify[12], a fact-checking website dedicated to verifying posts on Weibo, from April 2014 to November 2018 and 1,806 fake news pieces that were officially verified by the Weibo Community Management Center within the same timeframe. Using a similar approach, Yang et al. [34] constructed the *CHECKED* dataset, concentrating on detecting COVID-19-related fake news on Weibo. Additionally, Nan et al. [15] constructed the *Weibo-21* dataset, the first multi-domain Chinese fake news detection dataset. *Weibo-21* contains both fake and real news pieces gathered from Weibo spanning from December 2014 to March 2021, covering nine different domains, such as Science, Military, and Education. Most recently, Hu et al. [9] constructed a

---

[7]For convenience of presentation, when referring the values of macro F1 score or accuracy, we omit the percentage signs.

[8]The Weibo Community Management Center, Weibo's official fact-checking agency, exclusively examines news sourced from Weibo, whereas other fact-checking agencies a wide range of news sources. We hypothesize that the inclusion of additional fact-checking agencies will also expand the variety of news sources considered. Please refer to Table 2 for the full list of included fact-checking agencies.

[9]Our dataset and code are available at https://anonymous.4open.science/r/MCFEND-82DB.

[10]The websites for these social platforms are as follows. Twitter: https://www.twitter.com/; Facebook: https://www.facebook.com/; BuzzFeed: https://www.buzzfeed.com/; PolitiFact: https://www.politifact.com/; and NewsGuard: https://www.newsguardtech.com.

[11]http://service.account.weibo.com

[12]https://www.newsverify.com/

multi-modal retrieval augmented dataset *MR2*. This dataset consists of two subsets from Weibo and Twitter, respectively, covering news with images and texts, and provides evidence retrieved from the Internet for both modalities.

**Table 1: Summary of Chinese fake news detection dataset. Please note that the *MR2* dataset comprises two subsets, one from Weibo (Chinese) and one from Twitter (English). The statistics presented in this table specifically pertain to its Weibo (Chinese) subset.**

| Dataset | #News | Text | Image | Social Context | News Source |
|---------|-------|------|-------|----------------|-------------|
| *Weibo-16* | 5,656 | ✓ | | ✓ | Weibo |
| *Weibo-Media* | 5,802 | ✓ | ✓ | ✓ | Weibo |
| *Weibo-20* | 6,362 | ✓ | ✓ | ✓ | Weibo |
| *Weibo-21* | 9,128 | ✓ | ✓ | ✓ | Weibo |
| *MR2* | 6,976 | ✓ | ✓ | ✓ | Weibo |
| MCFEND | 23,974 | ✓ | ✓ | ✓ | Multiple Sources |

It is evident that existing datasets for Chinese fake news detection rely heavily on Weibo. To this end, we constructed the pioneering multi-source Chinese fake news detection dataset, termed MCFEND, which contains a 23,974 real-world Chinese news pieces collected from multiple sources across three distinct categories. We compared and summarized the Chinese fake news detection datasets in Table 1.

## 3 MCFEND DATASET

In this section, we introduce the data collection process for our Chinese multi-source fake news detection dataset MCFEND. In addition, we also perform a data analysis with an aim to illustrate the differences between various sources.

### 3.1 Overview

The MCFEND dataset contains news verified by 14 fact-check agencies from a wide range of sources, such as messaging apps, social platforms, and traditional news outlets. The 14 fact-check agencies are categorized in three groups. Table 2 presents the full list of the included fact-checking agencies in different groups.[13] Specifically, the first group comprises five Chinese fact-checking agencies listed as active by Duke Reporters[14], along with nine Chinese fact-checking agencies manually verified by experts as active and authoritative. The second group covers three existing English fake news detction datasets. News from the Group 2 is collected through a carefully

---

[13]The websites for the 14 fact-checking agencies are as follows. China Internet Joint Rumor Refuting Platform: https://www.piyao.org.cn/; Tencent Jiaozhen: https://vp.fact.qq.com/; China Daily Factcheck: https://www.chinadaily.com.cn/china/factcheck/; Taiwan FactCheck Center: https://tfc-taiwan.org.tw/; MyGoPen: https://www.mygopen.com/; HKBU Factcheck: https://factcheck.hkbu.edu.hk/home/; HKU Annie Lab: https://annielab.org/; AFP Fact Check Asia: https://factcheck.afp.com/afp-asia/; Factcheck Lab: https://www.factchecklab.org/; Politifact: https://www.politifact.com/; Politifact: https://www.politifact.com/; Gossipcop: https://www.gossipcop.com/; BS Detector: https://github.com/selfagency/bs-detector; and FakeNewsCorpus: https://github.com/architapathak/FakeNewsCorpus.
[14]https://reporterslab.org/fact-checking/

designed cross-lingual identical news retrieval method. Detailed description for the methods is in Sec. 3.2.2. Group 3 contains only Weibo Community Management Center, for which we directly utilized news data from the *Webo-21* dataset [15].

**Table 2: List of covered fact-checking agencies in three groups.**

| Group | Fact-checking Agencies |
|-------|------------------------|
| Group 1 | China Internet Joint Rumor Refuting Platform |
| | Tencent Jiaozhen |
| | China Daily Factcheck |
| | Taiwan FactCheck Center |
| | MyGoPen |
| | HKBU Factcheck |
| | HKU Annie Lab |
| | AFP Fact Check Asia |
| | Factcheck Lab |
| Group 2 | Politifact |
| | Gossipcop |
| | BS Detector |
| | FakeNewsCorpus |
| Group 3 | Weibo Community Management Center |

We gathered a total of 23,974 news pieces, comprising 8,144 from 14 fact-checking agencies in Group 1, 6,702 from three available English fake news detection datasets, which covers four English news-based fact-checking agencies in Group 2, and 9,128 from Weibo Community Management Center in Group 3. Similar to the existing datasets [15, 24], we collect the following information of each piece of news in any group. 1) Multi-modal news content, including text, images, and the metadata, e.g., timestamps. 2) Multi-modal social context, including posts, comments, emojis, user profiles, and other metadata, e.g., like counts of comments. Table 3 demonstrates the detailed statistics of the resulting MCFEND dataset.

**Table 3: Statistics of the MCFEND dataset**

| Statistics | Group 1 | Group 2 | Group 3 | Overall |
|------------|---------|---------|---------|---------|
| #Total | 8,144 | 6,702 | 9,128 | 23,974 |
| #Fake | 7,486 | 5,741 | 4,488 | 17,715 |
| #Real | 473 | 9,61 | 4,640 | 6,074 |
| #user | 235,215 | 156,862 | 458,800 | 803,779 |
| #posts | 58,299 | 41,600 | 70,814 | 170,713 |
| #comments | 262,342 | 328,465 | 1,512,095 | 2,102,902 |
| Timeframe | Mar. 2015 - Mar. 2023 | Jan. 2015 - Mar. 2023 | Dec. 2010 - Mar. 2021 | Jan. 2015 - Mar. 2023 |

### 3.2 Dataset Construction

In this subsection, we present the process of constructing the dataset for each fact-checking agency group. Fig. 2 illustrates the entire process for the dataset construction.

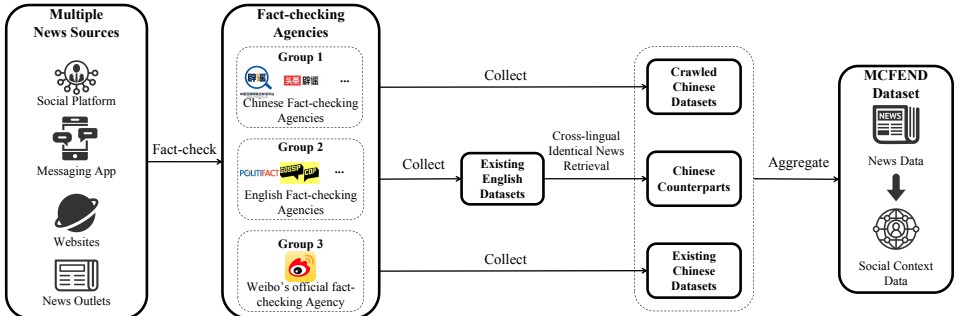

**Figure 2: The process for constructing the MCFEND dataset.**

*3.2.1 Group 1: Fact-checking Agencies Data Crawling.* Fact-checking agencies serve as a common source for labeling fake news detection datasets [8, 18]. These agencies are typically operated by government entities, companies, or non-profit organizations, and they employ authoritative experts to assess the authenticity of news pieces originating from diverse sources, such as social platforms, messaging apps, and traditional online news outlets. As discussed in Sec. 1, including a wider range of fact-checking agencies would also increase the diversity of news sources considered.

To maximize our coverage of news sources, we conducted web crawling to collect data from all active Chinese fact-checking agencies, encompassing the five Chinese fact-checking agencies identified as active by Duke Reporters, in addition to nine other Chinese fact-checking agencies that were manually verified as active and authoritative. In the case where the labels on some fact-checking agencies, e.g, AFP Fact Check Asia and Factcheck Lab, are presented in the form of images, we utilized optical character recognition method called Tesseract-OCR[15] to retrieve such labels.

*3.2.2 Group 2: Cross-lingual Identical News Retrieval.* To further diversify our news sources, we introduce a cross-lingual identical news retrieval method. This method searches for the corresponding Chinese counterparts of English real and fake news within annotated English datasets. Specifically, we utilized three widely used datasets: *FakeNewsNet* [24], *KaggleFakeNews* [21], and *FakeNewsCorpus* [16]. *FakeNewsNet* comprises two comprehensive subsets obtained from different fact-checking agencies, namely Politifact and Gossipcop. *KaggleFakeNews* contains news gathered from 244 sources classified as "unreliable or otherwise questionable" by the BS Detector, a browser extension that assesses web page links for reliability by comparing them to a professionally curated list. *FakeNewsCorpus* is a dataset consisting of news related to the 2016 US elections. The news pieces in this dataset are manually annotated by its authors. We consider the BS Detector and the authors of the *FakeNewsCorpus* as two distinct fact-checking agencies. By incorporating these three datasets, we effectively introduce data from four additional fact-checking agencies, enabling us to collect news from a wider range of English news sources.

For each news piece in these datasets, we executed the following steps to identify its corresponding Chinese counterpart.

- Step 1: Translation. We utilized the Baidu translation API[16] to translate the headlines of the English news into Chinese.
- Step 2: Chinese News Retrieval with Google News.[17] Google News provides extensive and up-to-date news coverage from sources worldwide. We configured the language and region of interest as "Chinese (China)" and employed the translated Chinese news headline as the search query. Search engines typically sort results by relevance. We assumed that the top five returned news pieces were the most relevant Chinese counterparts to the original English news. Subsequently, we crawled the top five returned news pieces.
- Step 3: Cross-lingual News Similarity Calculation. To determine the degree of similarity between the Chinese news retrieved in the previous step and the original English news, we employed the winning model [33] from the SemEval-2022 Task 8 challenge [4] for cross-lingual news similarity calculation. Specifically, we calculated the similarity score between the retrieved Chinese news and the original English news. The Chinese news with the highest similarity score were preserved in our MCFEND dataset.
- Step 4: Label Assignment. The authenticity label of the original English news is used to label its Chinese Counterpart, that is, the Chinese news with the highest similarity score.

The method inherently retains human-written news content. In contrast to directly utilizing Chinese news content generated by machine translation, our approach avoids unnatural textual expression that could potentially introduce irrelevant noise to the models.

*3.2.3 Group 3: Weibo News Collection.* Group 3 contains only Weibo Community Management Center. For this group, we directly utilized news data from the *Webo-21* dataset [15], the largest Chinese fake news detection datasets on weibo.

*3.2.4 Social Context Collection.* In some cases, depending on solely news content may be inadequate for detecting fake news, because fake news content is often meticulously crafted to deceive the public. Social platforms offer an invaluable source of supplementary

---

[15]https://github.com/tesseract-ocr/tesseract

[16]https://fanyi-api.baidu.com/
[17]https://news.google.com/

information in the form of social context features [8, 14, 23, 24, 26], which capture user interactions and social behaviors within the social platform environment. Thus, to incorporate such important features, we collected social context data, such as posts, comments, user profiles, etc., on the largest social platform in China, Weibo[18].

The process of collecting social context closely aligns with the approach used by *FakeNewsNet* for gathering social context from Twitter. Firstly, for news pieces that have headlines, we create search queries for associated posts on Weibo from the headlines. For news pieces without headlines, we utilize the jieba tool[19] to tokenize the textual content of the news and extract the top five keywords. These extracted keywords are then used as search queries. During this process, we remove special characters from the search queries to eliminate unnecessary noise. Then, we retrieve user responses to these posts, which include comments, reposts and likes. Additionally, when we identify all the users involved in the news propagation process, we collect metadata of these users, such as their usernames, and profiles.

As shown in Table 3, we assembled a comprehensive set of relevant social context data, which includes 170,713 posts and 2,102,902 comments from 803,779 distinct users.[20]

*3.2.5 Post-collection Processing.* After collecting all news pieces and their corresponding social context, we conducted three post-collection processing steps.

- Step 1: Text Cleaning. To enhance the data quality and eliminate unnecessary noise, we conducted text cleaning on text within both news contents and social context. This cleaning process involved the removal of HTML tags, punctuation, white spaces, stop words, prefix headings. We also replaced emojis with corresponding words, and unified time format.
- Step 2: Deduplication. The raw data contained multiple duplications. As a result, we removes redundant news and social context data to avoid unnecessary repetitions.
- Step 3: Label Mapping. Different fact-checking agencies employ diverse fine-grained labels to express the degree of authenticity (e.g., true, mostly true, and inconclusive). To ensure the consistency, we designed a label mapping strategy to standardize the original labels. Please refer to the supplementary material in our GitHub repository for the details of our strategy.

## 3.3 Comparison of the Three Groups

Our MCFEND dataset gathers news from 14 fact-checking agencies, categorized into three distinct groups. As previously noted, fake news originating from diverse sources are different in both the content and the social context [1, 16, 18, 25, 31]. To quantify these differences between the three groups, we conduct analyses on these two aspects which are considered as the significant clues for fake news detection [8, 18].

About the content analysis, we employed a pretrained Chinese Sentence-BERT model [20] to generate text representations for

each news piece. Consequently, each news piece was represented as a 768-dimensional vector based on its textual information. This representation serves as an encompassing approximation of the textual features of the news. To visually depict the differences, we applied t-SNE [27] to reduce the dimensionality of these vectors to two. As shown in Fig. 3, the textual features of news pieces collected from fact-checking agencies are distinct. This result validates the significant difference in their textual characteristics.

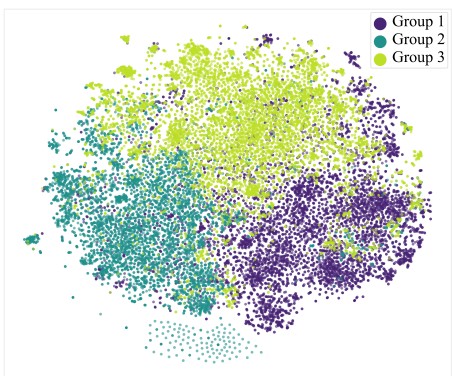

**Figure 3: Visualization of text representations for news collected from three distinct groups of fact-checking agencies.**

For the social context, we considered the significant social emotion feature [36], which is a 275-dimensional feature representing the emotions evoked within the social context surrounding the news pieces and has been verified to exhibit distinctions between fake and real news [36]. Similar to the content analysis above, we reduced the dimensionality of the social emotion feature to two. As illustrated in Fig. 4, the social emotion features associated with news pieces from Weibo demonstrate a more scattered pattern, whereas those from other sources are clustered. These results illustrate the substantial differences in their social context characteristics.

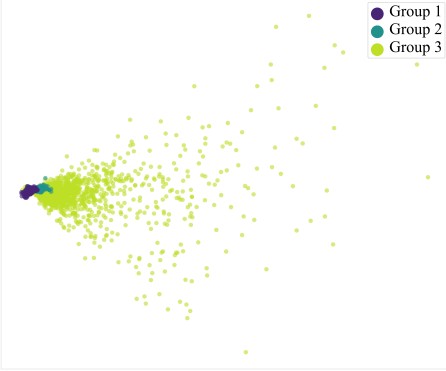

**Figure 4: Visualization of social emotion feature for news collected from three distinct groups of fact-checking agencies.**

In summary, our analysis of both content and social context provides strong evidence of differences between the three groups. We

---

[18]Note that while Weibo serves as the social context source for all collected news pieces, it also serves as an independent news source.
[19]https://github.com/fxsjy/jieba
[20]Individual users may be engaged in the social context of news collected from fact-checking agencies across different groups.

also consider other factors like text length and dual emotions between the news publisher and associated users in the social context. Due to the space limit, we omit the analysis here.

## 4 EXPERIMENTS

We conducted experiments to evaluate the performance of the existing representative fake news detection methods on our newly proposed MCFEND dataset. Specifically, we mainly answer the following evaluation questions (EQs):

EQ1: Are the existing methods, which have demonstrated effectiveness on prior Weibo datasets, capable of maintaining their performance when applied to news collected from different sources?

EQ2: Can training with multi-source data contribute to enhancing the robustness of existing methods in detecting fake news in the real-world scenarios which involves multiple sources?

### 4.1 Baselines

To obtain comprehensive answers to our evaluation questions, we carefully chose eight models from both commonly accepted categories of fake news detection, content-based and propagation-based methods [1, 8, 9], to establish baseline benchmarks. Their details including the implementations on the MCFEND dataset are as follows.

*4.1.1 Content-based Methods.* Content-based methods rely on only the textual and visual contents of the news itself. We adopted the two representative types of the content-based models, *uni-modal models* and *multi-modal models*.

*Uni-modal models* focus on the textual content of the news. We used BERT [7] and RoBERTa [12] as contextualized encoders to encode the textual content. Then the representation of the "[CLS]" special token is used for prediction. The implementation of BERT and RoBERTa in this study is based on their respective Chinese base version, i.e., BERT-base-Chinese[21] and RoBERTa-wwm-Base.[22]

*Multi-modal models* encode both text and images in input news. We include two baselines: CLIP [19] and CAFE [5]. CLIP [19] is a pretrained model for images and text. We input the image and text of the news into CLIP to generate a joint representation by computing the dot product between the visual and textual representations. This joint representation is then utilized for predictions. We implement CLIP based on its Chinese version.[23] CAFE [5] is an ambiguity-aware multi-modal fake news detection method. Specifically, it effectively combines uni-modal features and cross-modal correlations. It relies on uni-modal features when cross-modal ambiguity is weak and utilizes cross-modal correlations when cross-modal ambiguity is strong. CAFE demonstrates superior fake news detection performance on *Twitter* [2] and *Weibo-16* [10] datasets. It stands as the state-of-the-art content-based approach in this task.

*4.1.2 Propagation-based Methods.* Propagation-based methods are typically categorized into three groups: tree-based, modal fusion-based, and graph-based [9]. However, considering the nature of our MCFEND dataset, which contains news from diverse sources and lacks

the necessary cross-source user/news interactions to build effective graph neural networks, we included only the *tree-based models* and *modal fusion-based models* and excluded the graph-based methods in our study.

*Tree-based models* are designed to extract features from news and their associated comments using a tree structure, enabling them to capture complex propagation patterns. In our study, we included two competitive tree-based fake news detection models, namely Tree-RvNN [14] and Tree-Transformer [13], as our baseline models. Tree-RvNN [14] models the propagation of news using a top-down tree structure and generates feature vectors for news based on their propagation paths. It employs a Tree-LSTM [26] to directly encode the tree structure, followed by a max-pooling layer over the leaf comments to create the final representation. On the other hand, Tree-Transformer [13] leverages the attention mechanism to identify crucial comments within the social context and combines representations from both top-down and bottom-up trees.

*Modal fusion-based models* integrates information from both news content and its social context. In our study, we considered two baseline models for this category: dEFEND [23] and BERT-EMO [36]. The dEFEND model [23] utilizes a sentence-comment co-attention sub-network to exploit both news contents and comments in the social context to jointly capture explainable top-k checkworthy sentences and user comments for fake news detection. On the other hand, BERT-EMO [36] enhances a BERT-based fake news detector by incorporating dual emotion features that represent both dual emotions and the relationship between news and comments within the social context. Note that the BERT-EMO model has demonstrated outstanding performance in fake news detection, achieving the highest reported performance on the *Weibo-20* dataset [36]. Our preliminary experiments on the *Weibo-21* dataset [15] have also shown that BERT-EMO achieved an impressive F1-score of $98.30 \pm 0.005$, outperforming all other methods. The results establish BERT-EMO as the state-of-the-art propagation-based approach in this task.

### 4.2 Experimental Setup

Similar to existing study in fake news detection [9–11, 15, 36], we used accuracy and macro F1 as the evaluation metrics. The results are presented as mean values of F1 and accuracy with standard deviations across 5 runs for 100 epochs with a single NVIDIA GeForce RTX 3090 GPU. The Adam optimizer is used to train all the models. The batch size is set as 32. An early stopping strategy is adopted in the model training. We stop training the models if no improvement is identified after 500 consecutive batches which is enough to reach the convergence. For all textual content, we standardize them to a maximum length of 256 tokens; Regarding the images, we adopt a similar approach to that used in [9], which crops the center of each image to a size of 224 × 224 pixels. In the case where a news piece does not include an image, we provide a pure white image during the training of multi-modal models. The MCFEND dataset is divided into a train set, a validation set, and a test set, with a ratio of 7:1.5:1.5, respectively. The selection of hyper-parameters is based on the validation set. Please refer to the supplementary material in our GitHub repository for the details of hyper-parameters for all baselines.

---

[21]https://huggingface.co/bert-base-chinese
[22]https://huggingface.co/hfl/chinese-roberta-wwm-ext
[23]https://github.com/OFA-Sys/Chinese-CLIP

## 4.3 Cross-source Evaluation

To address EQ1, we performed cross-source evaluations on the selected baseline models. Specifically, these baseline systems were exclusively trained using news sourced from Weibo (i.e., data collected from the fact-checking agency in Group 3), and their corresponding social context in the train set of the MCFEND dataset.

**Table 4: The performance of the baselines in cross-source evaluation. The training data for all baselines in cross-source evaluation is exclusively sourced from Weibo. The highest accuracy or macro F1 score for each group is in bold text.**

| Model | Test Source | Accuracy | Macro F1 |
|---|---|---|---|
| BERT | Group 1 | 76.23±9.11 | **49.70±2.35** |
| | Group 2 | 76.93±7.05 | 50.84±5.02 |
| | Group 3 | 89.16±1.14 | 84.02±1.05 |
| | Overall | 74.74±1.57 | 70.97±1.46 |
| RoBERTa | Group 1 | 63.12±20.52 | 42.46±6.87 |
| | Group 2 | 54.34±29.92 | 54.34±29.92 |
| | Group 3 | 88.44±0.56 | 84.24±1.43 |
| | Overall | 76.47±3.65 | **72.97±2.91** |
| CLIP | Group 1 | **85.11±3.55** | 51.83±1.61 |
| | Group 2 | **82.87±3.59** | 56.90±2.21 |
| | Group 3 | 85.50±2.38 | 86.99±1.97 |
| | Overall | **84.48±2.61** | 65.08±0.59 |
| CAFE | Group 1 | 67.25±2.63 | 45.84±1.42 |
| | Group 2 | 35.47±4.41 | 33.36±3.46 |
| | Group 3 | 88.63±0.79 | 88.66±0.82 |
| | Overall | 63.78±1.81 | 61.03±1.40 |
| Tree-RvNN | Group 1 | 31.57±0.91 | 29.07±1.39 |
| | Group 2 | 26.24±3.54 | 24.28±3.20 |
| | Group 3 | 83.50±1.02 | 83.38±1.01 |
| | Overall | 47.94±1.57 | 47.61±1.64 |
| Tree-Transformer | Group 1 | 68.57±5.08 | 43.88±3.86 |
| | Group 2 | 49.52±4.09 | 41.88±3.14 |
| | Group 3 | 76.19±1.51 | 75.50±1.40 |
| | Overall | 64.76±2.59 | 58.93±1.54 |
| dEFEND | Group 1 | 68.69±9.83 | 47.79±2.96 |
| | Group 2 | 47.93±3.81 | 44.71±2.50 |
| | Group 3 | 74.74±2.11 | 72.38±2.45 |
| | Overall | 63.77±4.12 | 62.35±3.38 |
| BERT-EMO | Group 1 | 62.81±4.99 | 46.84±2.10 |
| | Group 2 | 65.83±2.45 | 47.55±1.20 |
| | Group 3 | **98.02±0.47** | **98.00±0.47** |
| | Overall | 75.56±2.39 | 70.41±2.03 |

The results of the cross-source evaluation are illustrated in Table 4. Notably, all baselines trained exclusively on data from the Weibo source exhibit a significant decrease in performance when applied to news sources other than Weibo (i.e., news sources covered by Group 1 and 2). These results highlight a critical observation that the baselines, including those existing representative ones, trained on existing Chinese fake news detection datasets, only composed of Weibo data, do not demonstrate robust performance in real-world scenarios where fake news originates from diverse sources. Therefore, our response to EQ1 is that the existing methods fail to

maintain their performance when applied to news collected from different sources. Besides, it is worth noting that all these baselines perform well when tested on Weibo-sourced data (Group 3). For instance, the state-of-the-art BERT-EMO model demonstrates impressive performance with a macro F1 score (resp. accuracy) of 98.00 (resp. 98.02) when tested on Weibo-sourced data. However, BERT-EMO's macro F1 score drops by 52.20% and 51.48% to 46.84 and 47.55 for Group 1 and Group 2, respectively, when applied to news whose sources other than Weibo. One contributing factor to such performance decrease is the significant difference in the social emotion feature, as shown in Fig. 4, between news sourced from Weibo and news from other sources. This discrepancy raises a crucial concern that the robustness of existing Chinese fake news detection methods is questionable, emphasizing the need for comprehensive re-evaluation before considering their practical deployment in the real-world scenarios.

**Table 5: The comparison of the average performance decrease of all categories of baselines. This decrease represents the difference between overall performance and performance on data sourced from Weibo. A smaller performance decrease indicates greater cross-source performance robustness.**

| Baseline Category | Average Macro F1 Decrease |
|---|---|
| *Uni-modal* | **-12.16** |
| *Multi-modal* | -24.77 |
| *Tree-based* | -26.17 |
| *Modal fusion-based* | -18.81 |
| Average | -20.48 |

Table 5 compares the average macro F1 score decrease, i.e., the difference between the overall performance and the performance on data sourced from Weibo, across all categories of baselines. Our experimental results show that the *uni-modal models*, which rely on solely the textual data in news content for fake news detection, has the best cross-source performance. The uni-modal model, RoBERTa, achieves the best cross-source performance. This suggests that variations in textual features across different sources may have a relatively modest impact on the performance of fake news detection models. In contrast, variations in visual and social context-related patterns in other sources have significantly harm models' performance. Mitigating the effects of variations in visual and social context-related patterns across different news sources is one interesting future direction to explore.

## 4.4 Multi-source Evaluation

To address EQ2, we performed multi-source evaluations on the selected baseline models. Specifically, these baseline systems were trained with the complete train set of the MCFEND dataset, including news from all sources covered in our dataset and their corresponding social contexts.

The results of the multi-source evaluation are presented in Table 6, which reveal several findings as follows. First and foremost, comparing the analysis from the cross-source evaluation (see Table 4), we can observe a substantial improvement in the performance of all baseline models after including multi-source data for training. Notably, the most significant improvement is observed in the

performance of CAFE, with an increase in its accuracy and macro F1 scores by 36.68% and 34.29%, respectively. To offer a qualitative analysis of the enhancement from using the multi-source data in the training process, we take the CAFE model as an example for illustration. When trained with the Weibo data exclusively, the CAFE model is unable to correctly identify the news pieces (c) and (d) shown in Fig. 1 as fake. However, when trained with data from all the diverse sources encompassed in the train set of MCFEND, the CAFE model exhibits the ability to accurately detect all of the presented fake news in Fig. 1. This results underscore the necessity of incorporating multi-source data for fake news detection models in the real world, which is beneficial to substantial enhancements in model performance and robustness.

Moreover, an interesting observation is that three baselines, i.e., BERT, CLIP, and Tree-Transformer, demonstrate enhanced performance on Weibo data after the inclusion of data from other sources

**Table 6: The performance of the baselines in multi-source evaluation. The training data for all baselines in multi-source evaluation contains news from all sources. The highest number in each group is in bold.**

| Model | Test Source | Accuracy | Macro F1 |
|---|---|---|---|
| BERT | Group 1 | 92.12±0.57 | 58.87±1.83 |
| | Group 2 | 89.65±0.51 | 77.69±0.94 |
| | Group 3 | 84.97±1.63 | 84.97±1.63 |
| | Overall | 89.30±1.18 | 84.50±2.17 |
| RoBERTa | Group 1 | 93.14±0.41 | 57.00±3.51 |
| | Group 2 | 89.51±0.46 | 74.37±4.46 |
| | Group 3 | 82.75±2.27 | 82.67±2.34 |
| | Overall | 88.12±0.68 | 84.38±0.45 |
| CLIP | Group 1 | **94.08±1.0**2 | 60.10±2.36 |
| | Group 2 | **90.86±0.98** | 78.64±3.26 |
| | Group 3 | 88.14±0.86 | 88.16±0.81 |
| | Overall | **90.88±0.65** | **86.23±0.49** |
| CAFE | Group 1 | 88.09±0.37 | **83.74±1.10** |
| | Group 2 | 88.20±0.20 | **83.15±0.61** |
| | Group 3 | 85.58±1.05 | 79.29±1.23 |
| | Overall | 87.17±0.34 | 81.93±0.73 |
| Tree-RvNN | Group 1 | 54.54±0.32 | 37.61±0.16 |
| | Group 2 | 58.53±2.77 | 36.90±1.11 |
| | Group 3 | 71.74±0.33 | 71.72±0.33 |
| | Overall | 64.71±0.36 | 61.04±0.30 |
| Tree-Transformer | Group 1 | 85.70±0.83 | 66.59±1.94 |
| | Group 2 | 76.05±0.99 | 59.74±0.56 |
| | Group 3 | 76.20±0.68 | 76.10±0.69 |
| | Overall | 79.31±0.59 | 73.13±0.79 |
| dEFEND | Group 1 | 90.58±0.39 | 58.03±1.34 |
| | Group 2 | 68.66±4.71 | 60.18±2.80 |
| | Group 3 | 72.05±2.26 | 70.62±2.89 |
| | Overall | 76.91±2.61 | 73.95±1.17 |
| BERT-EMO | Group 1 | 93.14±0.43 | 75.14±1.34 |
| | Group 2 | 83.65±1.16 | 70.13±1.49 |
| | Group 3 | **93.11±0.99** | **92.94±1.03** |
| | Overall | 89.54±0.39 | 81.46±0.43 |

during their training. Their respective macro F1 scores increases by 0.95, 1.17, and 0.60 respectively. The result indicates the advantages of introducing MCFEND dataset to training fake news detection models. It not only enhances the models' performance in addressing fake news from various sources but may also improve the effectiveness of certain models when dealing with single-sourced data, such as Weibo. One plausible explanation for this improvement lies in the ability of multi-source data to introduce a more comprehensive set of fake news features. This enables the models to better capture subtle distinctions between fake and real news across various sources. Additionally, the inclusion of multi-source data may serve as a kind of regularization, preventing the models from overfitting to the characteristics of the training data solely from the Weibo domain. Our finding highlight the potential of our proposed MCFEND dataset to serve as a valuable resource for improving Chinese fake news detection across both diverse and specific news sources.

Additionally, the overall performance of all baseline models when they were trained and tested on multi-source data is lower than their performance when they were trained and tested on Weibo data exclusively. Our findings address the challenge of developing algorithms capable of effectively distinguishing generic fake news features across news from various sources in the real-world scenarios. Among all the baseline models, the multi-modal model, CLIP, outperforms the others with an accuracy of 90.86 and a macro F1 score of 86.23. In contrast, the state-of-the-art content-based and propagation-based models, namely CAFE and BERT-EMO, exhibit suboptimal performance, achieving the macro F1 scores of 81.93 and 81.46, respectively. It's important to highlight that CLIP is a general multi-modal vision and language model, not specifically designed for the fake news detection task, whereas CAFE and BERT-EMO are custom-tailored for this particular task. Thus, there is a need for further enhancements in the development of new fake news detection algorithms.

## 5 CONCLUSION

In this work, we introduced the first multi-source benchmark dataset for Chinese fake news detection termed MCFEND. Different from the existing Chinese fake news detection datasets that are based on one single news source (Weibo), to construct the MCFEND, we aggregated the collected news from multiple sources that were fact-checked by 14 authoritative fact-checking agencies. To test the applicability of existing methods, we conducted a systematic evaluation of eight representative fake news detection models, including the state-of-the-art ones, in both cross-source and multi-source scenarios. Our experimental results reveal that models trained on data sourced exclusively from Weibo can be hardly applicable to the real-world scenarios where fake news usually originate from diverse sources. We also found that incorporating multi-source data for the model training enhances the robustness of the existing fake news detection methods. Our proposed MCFEND aims to be a benchmark dataset for evaluating Chinese fake news detection methods in the real world as well as advancing new effective methods in this area. One interesting future direction is to address multi-source data conflicts in the context of Chinese fake news detection. Certain data repairing methods, e.g., AutoRepair [35], can be leveraged when designing effective algorithms for detecting Chinese fake news.

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
