# OpenReview forum: "MCFEND: A Multi-source Benchmark Dataset for Chinese Fake News Detection"
_ACM.org/TheWebConf/2024/Conference — TheWebConf24 Oral_

### Official Review · Reviewer_qgE5 · 2023-11-18

**Novelty:** 4
**Technical Quality:** 5

**Review:**

The significance of this work is that it addresses the limitations of current single-sourced Chinese fake news detection datasets. The existing datasets are limited to Weibo as the only source of both true and fake news. However, in the real world, fake news emerges from multiple sources, such as social platforms, messaging apps, traditional online news outlets, etc. This makes it difficult to train and evaluate models on existing datasets, as they do not capture the diversity of real-world fake news.

The pros of this work are that it constructs a new dataset called MCFEND, which contains 23,974 pieces of authoritatively verified Chinese news from 14 fact-checking agencies. This dataset is larger than existing datasets and includes news from a variety of sources. Additionally, the work conducts cross-source and multi-source evaluations on eight established baseline models for Chinese fake news detection. These evaluations show that models trained on existing datasets are not applicable in real-world scenarios. Incorporating multi-source data is necessary and results in substantial improvements in models’ robustness. Authors also release the code and dataset publicly for community's benefit.

The cons of this work are that it is limited to Chinese fake news detection. It would be interesting to see if the same results could be obtained for other languages (such as English). Additionally, the experiment setting for EQ2 may not be sufficient to evaluate the models' general fake news identification capability. The reason is that if a model is trained on diverse sources, it may be obvious that it will be able to identify fake news from those diverse sources in a better way. A more reasonable setting would be increasing the number of fake news sources gradually and evaluating the performance of unseen sources. This evaluation is currently lacking in experiments.

**Questions:**

I would like to see more experiments in a setting where the number of unique training fake-news sources are increased gradually and the model is evaluated on other unseen sources. Does the models' generalizability improve in such a setting? This is important to understand since that's the main motivation behind this work.

**Reviewer Confidence:**

3: The reviewer is confident but not certain that the evaluation is correct

**Scope:**

3: The work is somewhat relevant to the Web and to the track, and is of narrow interest to a sub-community

---

### Official Review · Reviewer_DFyZ · 2023-11-29

**Novelty:** 4
**Technical Quality:** 5

**Review:**

Pros:
The paper is well-written and easy to understand.
I think it is an important issue of fake news detection from multiple platforms, so the dataset is an important support.

Cons:
I think the concept of “source” is not clearly defined, does it refer to sources the news pieces come from or the fact-checking agencies?
There are some typos in the paper.

**Questions:**

1. In Table 3, you mentioned the data of Group 3 ranges from Dec. 2014 to Mar. 2021, but I found that Weibo21 ranges from Dec. 2014 to Mar. 2021, can you confirm again?

2. How do you think about the difference between multi-source fake news detection and multi-domain fake news detection? Do methods for multi-domain fake news detection work for multi-source scenarios?

3. Again for the concept of “source”, I think this concept refers to different platforms, such as  Messaging App, News Outlets, etc. (Figure 2). But it seems like you refer to the three groups as different sources in experiments, I think it is necessary to clarify it more clearly.

**Reviewer Confidence:**

3: The reviewer is confident but not certain that the evaluation is correct

**Scope:**

4: The work is relevant to the Web and to the track, and is of broad interest to the community

---

### Official Review · Reviewer_NBj5 · 2023-11-30

**Novelty:** 4
**Technical Quality:** 6

**Review:**

**Summary**

The authors introduce a Multi-source benchmark dataset for Chinese FakE News Detection (MCFEND). While Chinese fake news detection has been relatively well-studied, most of the prevailing datasets originate solely from Weibo. Hence, the introduction of a multi-source benchmark dataset offers a diverse perspective, enhancing the assessment of trained fake news detection models. This dataset comprises 23,974 pieces of Chinese news, constructed using different methods from 14 fact-checking agencies.

**Strengths**

1. **Multi-Source Dataset**: The MCFEND dataset addresses an existing research gap where Chinese fake news models are exclusively trained and evaluated using data collected solely from Weibo. These single-source datasets pose a high risk to the efficacy and generalizability of the models.
2. **Innovative Dataset Construction Approach**. The authors employed an innovative technique to obtain the second group within the MCFEND dataset through cross-lingual identical news retrieval. This method proves effective in gathering credible misinformation instances, thereby expanding the scope of hate speech beyond what may be encompassed on Chinese platforms.
3. **Comprehensive Experiments**. The authors systematically conducted experiments on an extensive array of baseline models, encompassing content-based methods, propagation-based approaches, and others. The detailed experimental settings enhance comprehension of the various models using Chinese fake news datasets. Additionally, the authors conducted a thorough evaluation on the distinct groups within the MCFEND dataset, further underscoring the significance of a multi-domain dataset.

**Weakness**

1. **Writing contradictions.** In the introduction, the authors stated that the construction of the first group involves fabricated news data gathered from nine fact-checking agencies. However, a contradiction arises in the MCFEND dataset construction section, where the authors specify that the first group comprises five Chinese fact-checking agencies identified as active by Duke Reporters, in addition to nine other Chinese fact-checking agencies.
2. **Selection of posts from search queries**. While the authors mentioned the process of retrieving posts from Weibo (i.e., via headlines or extracted keywords) in Section 3.2.4 Social Context Collection, it is not clear how the authors determine the relevant posts and the point of stoppage.
3. **Unusual Experiment Results**. In Table 4, although the authors emphasized that the BERT model achieved the highest Macro F1 score for Group 1, it is observed that CLIP actually has a higher Macro F1 score (51.83 versus 49.70). Furthermore, there is confusion regarding the authors' computation of the overall Macro F1 score. While the table indicates that the CLIP model consistently outperforms the RoBERTa model, the CLIP model surprisingly has a lower overall Macro F1 score. Similar inconsistencies can be noted in Table 6 (CAFE versus BERT-EMO).

I have read the author's rebuttal, and their comments have clarified many of the weaknesses surrounding the experiments.

**Questions:**

1. How many groups are used in constructing the first group in the MCFEND dataset? (related to Weakness #1)
2. What are the selection criteria for the posts obtained from the search queries? What is the point of stoppage? (related to Weakness #2)
3. How are the (overall) Macro F1 scores computed? If there is a misunderstanding, can you kindly guide me through the evaluation steps?
4. What is the length breakdown of the various posts? This is related to the experimental setup where the authors standardize the post to a maximum length of 256 tokens.
5. Is the original post content preserved? Since language models (i.e., BERT and RoBERTa) are pre-trained on a large corpus of English data in a self-supervised fashion, the text post-processing might hurt their performance instead.

**Reviewer Confidence:**

4: The reviewer is certain that the evaluation is correct and very familiar with the relevant literature

**Scope:**

4: The work is relevant to the Web and to the track, and is of broad interest to the community

---

### Official Review · Reviewer_CThV · 2023-12-01

**Novelty:** 6
**Technical Quality:** 6

**Review:**

This paper presents a new dataset for fake news detection in Chinese. The dataset is compiled from many different sources and contains 23k examples of multimedia content that was fact-checked by 14 fact-checking agencies. The authors then evaluate 8 fake news detection approaches on this dataset and show that models trained on commonly used Weibo-only content do not generalize to other sources and incorporating training data from different sources improves the accuracy of these models.

This is an interesting paper that makes a useful contribution to the detection of fake news in Chinese by creating a large, multi-source, multi-modal dataset.

The collection methodology is well-documented and the dataset greatly improves the ability of future researchers to study this phenomenon across different platforms.

The baselines chosen are fairly robust and the difference seen in their original performance at the task vs. that after being trained on this new dataset shows that it greatly improves their ability to detect fake news in the wild.

However, some things in the approach are unclear. In the fake news detection pipeline, the authors standardize the text to 256 tokens, which is much shorter than the average news article length and would remove much of the context needed to judge whether it is fake news or not.

It is also not clear whether any deduplication is performed since it could be that multiple instances of the same fake news story have been obtained from different sources, which would inflate the accuracy of the models if they have already seen similar samples in the training set.

It would be helpful to see some descriptive analysis of the difference between fake news obtained from different sources in addition to the embedding projections presented. Factors like average length, topic distribution, linguistic style, amount of images included, would be helpful in understanding the nature of fake news across the different platforms.

The authors may wish to try using UMAP instead of t-SNE for dimensionality reduction for the plots in Figure 3 and 4 as it can preserve global structure better. It would also be good to see some annotation of different topics within the plot.

Overall, this is a well-written paper and the figures and tables are of good quality as well.

---

I have read and replied to the authors' response and made the necessary changes to my review.

**Questions:**

What was the accuracy of the OCR used to extract fact checking labels?

How was it verified that the Chinese articles found via translation do indeed contain the same misleading statements as the original, and not just neutral coverage of an English fake news article?

Sentence-BERT has a limited context length, how were longer pieces of text truncated or chunked in order to generate their vector representations in Section 3.3?

Typos and presentation improvements:

In Section 1, the footnote number for footnote 9 in the text bleeds outside the column margins.

In Section 3.1, the footnote with the fact checking website URLs could be provided as a list or table in the appendix for better readability.

**Reviewer Confidence:**

3: The reviewer is confident but not certain that the evaluation is correct

**Scope:**

4: The work is relevant to the Web and to the track, and is of broad interest to the community

---

### Decision · Program_Chairs · 2024-01-22

**Decision:**

Accept (Oral)

**Comment:**

This is the meta-review by the SPC responsible for your paper, and takes into account the opinions expressed by the referees, the subsequent decision thread, and my own opinions about your work.

 - This paper presents a large multi-source Chinese fake news detection dataset called MCFEND. Unlike existing narrative datasets collected solely from Weibo, the authors employed an effective and innovative technique to collect fake news from multiple sources.
 - Overall, reviewers recognized the value of this work in advancing Chinese fake news detection, and the experiments are comprehensive and robust.
 - However, reviewers expressed some concerns, which the authors somewhat addressed during the rebuttal stage and committed to resolving in the final version.